# Comparative Transcriptome Analysis Reveals Regulatory Networks during the Maize Ear Shank Elongation Process

**DOI:** 10.3390/ijms22137029

**Published:** 2021-06-29

**Authors:** Cai-Yun Xiong, Qing-You Gong, Hu Pei, Chang-Jian Liao, Rui-Chun Yang, Gao-Ke Li, Jun Huang

**Affiliations:** 1Guangdong Provincial Key Laboratory of Plant Molecular Breeding, South China Agricultural University, Guangzhou 510642, China; xiongcy@stu.scau.edu.cn (C.-Y.X.); rchyang@scau.edu.cn (R.-C.Y.); 2Zhuhai Modern Agriculture Development Center, Zhuhai 519070, China; gongqingyou@163.com; 3College of Agriculture, South China Agricultural University, Guangzhou 510642, China; peihu98@163.com; 4Technical Research Center of Dry Crop Variety Breeding in Fujian Province, Crop Research Institute, Fujian Academy of Agricultural Sciences, Fuzhou 350013, China; liaocj1978@163.com; 5Guangdong Provincial Key Laboratory of Crop Genetic Improvement, Crop Research Institute, Guangdong Academy of Agricultural Sciences, Guangzhou 510640, China

**Keywords:** maize, ear shank length, transcriptome, regulatory network

## Abstract

In maize, the ear shank is a short branch that connects the ear to the stalk. The length of the ear shank mainly affects the transportation of photosynthetic products to the ear, and also influences the dehydration of the grain by adjusting the tightness of the husks. However, the molecular mechanisms of maize shank elongation have rarely been described. It has been reported that the maize ear shank length is a quantitative trait, but its genetic basis is still unclear. In this study, RNA-seq was performed to explore the transcriptional dynamics and determine the key genes involved in maize shank elongation at four different developmental stages. A total of 8145 differentially expressed genes (DEGs) were identified, including 729 transcription factors (TFs). Some important genes which participate in shank elongation were detected via function annotation and temporal expression pattern analyses, including genes related to signal transduction hormones (auxin, brassinosteroids, gibberellin, etc.), xyloglucan and xyloglucan xyloglucosyl transferase, and transcription factor families. The results provide insights into the genetic architecture of maize ear shanks and developing new varieties with ideal ear shank lengths, enabling adjustments for mechanized harvesting in the future.

## 1. Introduction

Maize (*Zea mays L.*) is a staple crop all over the world; it provides more than 30% of the food supply [1], and is also a significant resource for feed and biofuel. The lack of corn varieties with low grain moisture at harvesting stage affects mechanized harvesting in China. Therefore, the cost of maize production remains high. Maize ears are composed of different parts such as grains, husks, cobs and shanks [2,3]. The ear shank length has two main effects: photosynthetic products of the leaves are transported to the kernels through the ear shanks, thereby increasing the yield and quality [4]; additionally, the shank length affects the tightness of the husks, thereby affecting the dehydration of the ears. An excessive shank length may result in more husks on the ear during the mechanical picking of corn, which makes the peeling roller more likely to damage the corn kernels [5,6,7]. Therefore, breeding maize with an appropriate shank length is conducive to a satisfactory grain yield and mechanized harvesting.

Previous studies on maize ear shanks have mainly focused on the role of organic matter transportation. The vascular bundle in the shank represents the end of the long-distance transportation of sucrose for ear growth [8]. The ability of ear shanks to transport organic matter is affected by environmental factors and specific treatments. Water deficit will reduce the number of vascular bundles in the ear shank and the cross-sectional area of the shank. The decrease in the number of vascular bundles in the shank will significantly reduce the thousand-kernel weight, although the decrease in cross-sectional area has almost no effect on the kernel [8]. The application of appropriate concentrations of gibberellins (GA_4+7_) to maize shanks can increase the content of plant hormones such as auxin, gibberellin (GA), and abscisic acid (ABA), which increase the grain filling rate and postpone the senescence of the husks, resulting in an increase in maize grain yield [9]. Under moderate drought, exogenous ABA applied in field-grown maize stimulates greater production by increasing the number of vascular bundles and the area of phloem and enhancing the migration of carbohydrates to the grain [10]. These studies only clarify the function of ear shanks; however, they did not explore the genetic basis from the molecular level.

There have been few studies examining the genetic basis of maize shank elongation. It has been established that the shank length is a quantitative trait. In an F_2_ population from two inbred lines with long and short shanks, it was observed that the shank length showed extremely significant additive effects [4,11]. In the jasmonic acid (JA)-deficient *opr7*/*opr8* double mutant, *opr7* and *opr8* lost the ability to inhibit shank elongation, resulting in extreme ear shank lengths; this suggests that JA is necessary for inhibiting the elongation of the shank [12]. Shank length is controlled by multiple genes with small effects which are involved in hormone metabolism and sphingolipid biosynthesis; the regulation mechanism of shank length is complicated and not completely the same in diverse maize populations [4].

Transcriptome sequencing is widely used in contemporary maize research. Using RNA-sequencing technology, Shi et al. [13] discovered the early shade response mechanism of inbred B73-line seedlings, involving light signal transduction, auxin responses, and cell elongation pathways. Ma et al. [14] detected seven candidate genes associated with element uptake and utilization, hormone responses, and transcription factors, regulating maize root growth when suffering from nutrient deficiencies. Transcriptome analysis is also used to determine candidate resistance genes. Kebede et al. [15] identified differentially expressed genes located within quantitative trait locus (QTL) regions for resistance to gibberellin ear rot (GER), exploring cell detoxification, hormone transduction, and the biosynthesis of pathogenesis-related proteins and phytoalexins. Nevertheless, transcriptome sequencing has never been used to explore the mechanism of maize shank elongation.

The genetic basis and regulatory mechanisms of maize ear shank elongation remain unclear. Here, we identified differentially expressed genes (DEGs) via transcriptome sequencing analysis during the ear shank elongation process, analyzing the metabolic pathways involved. The results can provide insights for research into the genetic basis of maize ear shanks and the development of new varieties with ideal ear shank lengths for mechanized harvesting in the future.

## 2. Results

### 2.1. Phenotype Analysis of Maize Ear Shanks at Different Developing Stages

In order to study the regulatory network during the maize ear shank elongation process, we selected four developing stages with significant lengths of maize ear shanks for transcriptome sequencing (Figure 1A–D), corresponding to L1 (1.01 ± 0.02 cm), L2 (1.98 ± 0.03 cm), L3 (3.02 ± 0.04 cm) and L4 (3.98 ± 0.03 cm). Cytological analysis indicated that the lengths and widths of maize ear shank cells in different developing process (L1 to L4) were significantly different (Figure 1E–H,J,K), which revealed that shank elongation is accompanied by the growth of the cells.

### 2.2. Temporal Expression Patterns of Genes

In this study, eight libraries were constructed for RNA sequencing at different developmental stages (L1~L4) (with two biological replicates for each stage). The detailed sequencing data are shown in Appendix A. After quality control, clean read form samples ranged from 19.55 M to 24.71 M; Q30 (%) were 93.81 to 94.77. After mapping these clean reads to the B73 reference genome (version 4), over 168.85 M reads were mapped to the B73 genome and used for gene expression analysis. The “Unique Mapped Reads” ranged from 79.48% to 89.27%, and “Multiple Mapped Reads” ranged from 2.48% to 2.96%.

We identified 31,529 genes expressed during the maize ear shank elongation process; gene expression profile clustering was performed using Mfuzz function for time series analysis by http://www.bioinformatics.com.cn, an online platform for data analysis and visualization, accessed on 20 June 2021. The expression patterns of all 31,529 genes were clustered into nine profiles (Figure 2 and Appendix A). In the maize ear shanks, cluster 3 and cluster 7 showed similar expression patterns, with a peak expression which dropped thereafter at the L3 stage. Both cluster 2 and cluster 9 exhibited up-regulation at the late (L4) stage of the ear shank growth process. The expressions of cluster 1 and cluster 4 decreased significantly at the L2 stage; in contrast, cluster 5 and cluster 8 showed up-regulation at this stage. The genes in clusters 1, 3, and 7 exhibited an increasing trend at L3, but clusters 5, 6, and 8 decreased significantly at the L3 stage. Notably, the expression levels of cluster 2 and cluster 9 were lower at the L1, L2 and L3 stages, but increased significantly at the L4 stage. The expression level of cluster 5 was high at the L1, L2 and L3 stages, but decreased significantly at the later stage.

Gene Ontology (GO) classification analysis and Kyoto Encyclopedia of Genes and Genomes (KEGG) pathway analysis were performed to identify the putative functions. We analyzed the top 20 GO terms and top 20 pathways enriched by each cluster; significantly enriched GO terms and pathways were used to draw heatmaps (Appendix A). Interestingly, the “organonitrogen compound metabolic process” was significantly enriched in three clusters (clusters 1, 4 and 9), and “Kinase activity” was significantly enriched in cluster 2 and cluster 8. Compared with other clusters, cluster 4 enriched most GO terms, most of which were highly significantly enriched. We identified that the “Biosynthesis of amino acids” “Endocytosis”, “Arachidonic acid metabolism”, “Phagosome” and “Glycosylphosphatidylinositol (GPI)-anchor biosynthesis” pathways were significantly enriched in multiple clusters.

### 2.3. Functional Classification of DEGs

Based on gene expression profiles, differentially expressed genes were identified for different developing stages. There were 2684, 1113, and 6704 DEGs identified when comparing stages L1 vs. L2, L2 vs. L3, and L3 vs. L4, respectively (Figure 3A). We identified a total of 8145 DEGs during the shank elongation process. A Venn diagram was created to identify DEGs in all comparison sets (Figure 3B); a total of 246 DEGs were identified as differentially expressed in different developing processes.

To infer the biological processes of all DEGs, we conducted a GO classification analysis using the OmicShare tool, a free online platform for data analysis (GENE DENOVO, Guangzhou, China, http://www.omicshare.com/tools, accessed on 20 June 2021). The top 20 GO terms are displayed in Table 1; these 20 terms were mainly related to the enzyme activity, photosynthesis, and microtubules, such as “hydrolase activity, acting on glycosyl bonds” (GO:0016798), “kinase activity” (GO:0016301), “thylakoid part” (GO:0044436), “microtubule associated complex” (GO:0005875). A KEGG pathway analysis using a free online platform for data analysis (GENE DENOVO, Guangzhou, China, http://www.omicshare.com/tools, accessed on 20 June 2021) identified the significantly enriched 20 pathways in Table 2. Some pathways enriched in numerous genes had high significance, such as “ribosome” (ko03010), “biosynthesis of secondary metabolites” (ko01110), “metabolic pathways” (ko01100), “plant-pathogen interaction” (ko04626), “plant hormone signal transduction” (ko04075) and “phenylpropanoid biosynthesis” (ko00940). Similarly, photosynthesis-related pathways are also enriched, which is consistent with the GO results.

### 2.4. DEGs Involved in the Plant Hormone Signal Transduction Pathway

The “Plant hormone signal transduction” pathway which was significantly enriched is presented in Table 2. In our results, 96 DEGs were associated with plant hormone signal transduction, including ABA, brassinosteroids (BR), cytokinin (CTK), ethylene (ETH), JA, and salicylic acid (SA) signaling pathway. A total of 48 DEGs were mainly enriched in auxin signal transduction. There were seven DEGs in brassinosteroid signal transduction, and two DEGs in gibberellin signal transduction. An overview of some gene expression patterns during shank elongation is provided in Figure 4.

At the beginning of the auxin signaling pathway, the expression patterns of *TIR1* and *AUX1* were exactly opposite, and the levels of up-regulation and down-regulation were both significant. Twelve *AUX/IAA* and three *CH3* were highly expressed in L1. Seven and eight *SAUR* were up-regulated in L1 and L3, respectively. In the gibberellin and brassinosteroid signal transduction pathways, the differential genes had almost no similar expression patterns. In general, the expression patterns of DEGs enriched in “plant hormone signal transduction” were diverse, which provided a guarantee for elongation of the shank. This suggested that the hormone regulation of shank elongation itself is a considerably complicated process.

### 2.5. DEGs Involved in the Xyloglucan Metabolic Process and Xyloglucan Xyloglucosyl Transferase Activity

In cells, xyloglucan has a structural role, can interact with cellulose, and is important in cell wall extension and cell expansion [16]. Xyloglucan xyloglucosyl transferases are cell wall-modifying enzymes that play a fundamental role in expansion and remodeling, are differentially expressed in tissues, and are time- and space-dependent during the plant developing process [17]. In our study, we identified a total of 25 DEGs involved in the xyloglucan metabolic process and 16 DEGs involved in xyloglucan xyloglucosyl transferase activity; these are presented in Appendix A. We have provided an overview of gene expression patterns during shank elongation in Figure 5A,B.

In the process of xyloglucan metabolism, we found that some genes were up-regulated in multiple stages. There were also some genes that were only up-regulated in one stage and down-regulated in other stages. In Figure 5B, we present six DEGs (*Zm00001d027313*, *Zm00001d024071*, *Zm00001d024382*, *Zm00001d014617*, *Zm00001d024386*, *Zm00001d029814*) which were enriched in xyloglucan xyloglucosyl transferase activity, and were significantly higher expression in L1, but with a reduced expressed in the other three stages. Seven DEGs (*Zm00001d021667*, *Zm00001d051867*, *Zm00001d030103*, *Zm00001d015977*, *Zm00001d017699*, *Zm00001d047970*, *Zm00001d052651*) were significantly up-regulated in L3, and these genes were also up-regulated in other stages. This showed that these differentially expressed genes were regulated by different expression patterns in the process of shank elongation, which may play important roles in the shank elongation process.

### 2.6. DEGs Involved in Steroid Biosynthetic Process

In this study, we also identified 12 DEGs related to the “steroid biosynthetic process”. Brassinosteroid is a kind of steroidal plant hormone, and is an essential regulator in plant growth and development [18]. Here, we speculate that these enriched differentially expressed genes may be related to the synthesis of brassinosteroids. The functional annotations of these genes showed that they are important enzymes for the synthesis of steroids (Appendix A). We have also provided an overview of gene expression patterns, in Figure 5C. Seven (*Zm00001d009666*, *Zm00001d019669*, *Zm00001d039089*, *Zm00001d047797*, *Zm00001d028653*, *Zm00001d040173*, *Zm00001d050409*) and four (*Zm00001d028654*, *Zm00001d013037*, *Zm00001d037784*, *Zm00001d045078*) genes were significantly up-regulated in stage L1 and L2, respectively. There was only one gene (*Zm00001d044104*) highly expressed in stage L3.

### 2.7. Expression of TFs Involved in the Phytohormone Response Factor

Transcription factors can effectively regulate secondary metabolite production. Here, maize TF gene family information was obtained from the Plant Transcription Factor Database (PlantTFDB 2.0, Peking University, Beijing, China, http://planttfdb.cbi.pku.edu.cn/, accessed on 21 April 2021) [19]. Further analysis of our transcriptome data revealed that 729 genes were annotated to encode putative TFs, belonging to 45 families (Figure 6A). The top five largest of the 45 transcription factor families identified were MBY (85), bHLH (59), bZIP (52), NAC (52), and AP2 (46). In addition, we detected Aux/IAA (24), ARF (12), ERF (7), and GRF (5) transcription factor families directly related to phytohormonal responses which were differentially expressed. The expression dynamics of the TF genes revealed that these TF families had different expression patterns during maize ear shank elongation (Figure 6B).

Notably, almost all GRF genes were expressed at lower levels in stages L1 and L2; three genes (*GRF9*, *GRF1*, *GRF13*) were highly expressed in stage L4; and the other two genes (*GRF10*, *GRF6*) were poorly expressed in L4. Almost all AUX/IAA genes were highly expressed in stage L1, and some genes were highly expressed in more than two stages. In general, the AUX/IAA family maintained a high level of expression throughout the developing process. The ARF family was significantly up-regulated in all four stages. Almost all of the ERF family was highly expressed in stage L3. There were also some genes up-regulated in the L1 or L4 stages. This indicated that the hormone-related transcription factor families had a complicated mechanism for regulating ear shank elongation.

### 2.8. Validation of DEGs by qRT-PCR

The expression levels of the DEGs from transcriptomic sequencing were verified by quantitative RT-PCR. Nine DEGs, including *Aux/IAA-transcription factor 22* (*Zm00001d013707*), *ARF-transcription factor 23* (*Zm00001d038698*) and *Aux/IAA-transcription factor 10* (*Zm00001d051911*) were randomly selected to validate the RNA-seq results (Figure 7). The expressions of the nine DEGs were basically consistent with the ratios of the RNA-seq, confirming the reproducibility of the data.

## 3. Discussion

The maize ear shank, a unique channel that transports photosynthetic products to the ear, has two main effects. Firstly, the ear shank length is negatively correlated with maize yield; secondly, ear shank length is significantly correlated with husk number and length, affecting the dehydration rate of maize by reducing the air permeability between the husk and grains [4]. Therefore, breeding maize varieties with an appropriate shank length is very important for maize production and the development of mechanized harvesting. However, the genetic basis of maize ear shank elongation remains unclear. The purpose of this study was to identify candidate genes associated with maize ear shank elongation and investigate the potential regulatory networks. The comparative transcriptome analysis of tissues at different developing stages can provide valuable information on how the regulatory gene network controls specific development processes [20]. In this study, we investigated the gene expression patterns and networks at four developing stages to further understand the molecular mechanisms of the maize ear shank length elongation process.

Histological analysis revealed that ear shank elongation is related to cell division; the RNA-seq data from different development stages were used to detect putative important genes in the shank elongation process. In this study, 2684, 1113, and 6704 DEGs were identified in L1 vs. L2, L2 vs. L3, and L3 vs. L4, respectively. A total of 8145 DEGs were determined to be differentially expressed in different stages. Gene annotation enrichment analysis revealed that hormone signal transduction, xyloglucan metabolic and transcription factors may play important roles in ear shank development.

Phytohormones, as physiological signals that regulate plant growth and development, are very important for plant growth and development. Plant cell elongation is regulated by several plant hormones, including auxin [21,22,23], gibberellin [23,24,25,26], and brassinosteroids [25,27,28]. Networks of interconnected signal transduction pathways coordinately combine hormones and environmental signals (light, temperature) to express common cell activities and developmental processes [29,30]. Auxin plays a major role in regulating cell expansion by activations of cell wall synthesis and modification-related genes; these genes may be related to acid growth and the action of loose proteins, stimulating cell elongation by increasing the wall extensibility (such as wall loosening) [31]. Gibberellin (GA) promotes stem elongation by increasing cell division and cell expansion [23]. GA deficiency is exhibited as a dwarf phenotype; for instance, under plant growth regulator paclobutrazol treatment, the activity of ent-kaurene oxidase (KO) is inhibited which hinders the biosynthesis pathway of GAs [32]. BRs are also important positive regulators of stem elongation, and mutants with defects in biosynthesis or signal transduction show semi-dwarfing [33]. Auxin–GA cross-talk is important; it has been confirmed that auxin moves into the elongating internodes and maintains *PsGA3ox1*, and consequently mediates GA_1_ biosynthesis to promote garden pea shoot elongation [34]. BR was also quickly induced by auxin, and can promote stem elongation and inhibit root elongation in various plant species [35]. Under flooding conditions, BR can induce GA signals to regulate rice stalk elongation [36]. In our study, we identified that genes involved in auxin, brassinosteroid, and gibberellin signal transduction were differential expressed at different elongation stages (Figure 4), and these genes may be important candidate genes associated with maize ear shank elongation.

In plants, internode elongation depends on the growth of longitudinal cell elongation, and the number of cells increases through cell division [37]. Additionally, cell division is accompanied by the irreversible extension of the cell wall under the control of a network structure composed of polysaccharides such as cellulose and xyloglucan [38,39]. Here, we identified 25 DEGs involved in the xyloglucan metabolic process. Xyloglucan is a hemicellulose and occurs widely in the primary cell wall; it determines the tension of the cell tissue. During cell expansion and elongation, the primary wall, which confines and shapes the cell, is selectively loosened [39]. Plant xyloglucan xyloglucosyl transferase, classified as glycoside hydrolase family 16 (GH16), constitutes cell wall modification enzymes, which play a basic role in cell wall expansion and remodeling [17,40]. In this study, we identified 16 DEGs involved in xyloglucan xyloglucosyl transferase activity. Xyloglucan xyloglucosyl transferase can cut and reconnect one xyloglucan chain to the non-reducing end of another xyloglucan chain to regulate the rearrangement of polysaccharide chains during cell growth [41,42]. The growth of plant cells can be controlled by the molecular size of free xyloglucans and the integration mechanism of xyloglucan oligosaccharide accelerating cell wall elongation [43]. Both xyloglucan and xyloglucan xyloglucosyl transferase are essential for maintaining the dynamic structure of the cell wall, which occurs in plant cells, and can regulate balance between rigidity and flexibility. The flexibility of the cell wall allows the cell to expand, resulting in cell elongation.

Transcription factor families play an important role in regulating plant cell elongation and expansion. Our transcriptome data of maize ear shank elongation showed that 729 genes were annotated to encode putative TFs, belonging to 45 families. We identified some DEGs directly related to hormone-responsive factor transcription factor families, including Aux/IAA (24/39), ARF (12/37), ERF (7/16) and GRF (5/15) superfamilies. Furthermore, the top five largest of the 45 identified transcription factor families were MBY (85), bHLH (59), bZIP (52), NAC (52), and AP2 (46), as shown in Figure 6A. In plants, these transcription factor families are related to hormone signaling transduction during cell elongation. It has been proven that overexpressed *ZmMYB59* during seed germination promotes the synthesis of ABA and inhibits the synthesis of gibberellin and cytokines, thereby hindering cell elongation [44]. The application of exogenous brassinolide can up-regulate the expression of *GhbHLH/HLH* genes in cotton fiber. On the contrary, when brassinazole (Brz, a BR biosynthesis inhibitor) is applied, the expression of these *GhbHLH/HLH* genes is significantly down-regulated. This indicates that bHLH/HLH genes may participate in BR signaling transduction during cotton fiber development [45]. Basic-leucine zipper (bZIP) family can reportedly participate in GA-triggered cell elongation [46,47]. The maize bZIP transcription factor (*ZmGRF*), combined with the downstream target gene ear-kaurene oxidase (*AtKO1*), can promote flowering and cell expansion in Arabidopsis by acting as an activator of GA biosynthesis [47]. NAC transcription factors can directly inhibit the expression of key genes for gibberellin (GA) and brassinosteroid (BR) biosynthesis, leading to GA and BR defective phenotypes. NAC transcription factors can also reduce the expression of the bHLH transcription factor that positively controls cell elongation, while stimulating the expression of growth-suppressing genes [48]. The elongation and proliferation of rice root meristem cells is controlled by AP2/ERF transcription factor-mediated GA biosynthesis in a developmental stage-specific manner [49]. AP2 and JA may interact to inhibit the GA-mediated promoted pathway of stem elongation in the reproductive phase [36]. Figure 8 illustrates the relationship between the five largest transcription factor families which were enriched, as well as several hormones that control cell elongation.

## 4. Materials and Method

### 4.1. Plant Materials

In this study, inbred line B73 were planted in the Zengcheng Experimental Teaching Base of South China Agricultural University (Guangzhou, Guangdong, China) in autumn, 2020. To explore the shank elongation mechanism during the development process of maize ear shanks, the second internodes (the longest node) of the maize ear shank at four developing stages L1 (1.01 ± 0.02 cm), L2 (1.98 ± 0.03 cm), L3 (3.02 ± 0.04 cm) and L4 (3.98 ± 0.03 cm) were used for further analysis. For transcriptome sequencing, at each sampling point, two shanks were collected for RNA extraction. In total, eight libraries, corresponding to four developing stages, were used for RNA-seq.

For histological analysis, the second nodes of the shanks at different developing stages (L1–L4) were fixed in FAA (63% ethanol, 1.85% formaldehyde, 5% acetic acid in distilled water). Longitudinal sections of the stems were then cut using a double-edged razor and suspended in ddH_2_O. Images were observed using the microscopic imaging system of Olympus CKX53 apparatus (Olympus, Shinjuku, Japan, http://www.olympuscap.com/, accessed on 10 December 2020).

### 4.2. RNA Extraction and Library Preparation

The total RNA was extracted from the samples using a Plant Total RNA Purification Kit (TR02–150, GeneMarkbio) following the manufacturer’s instructions. RNA purity was checked using a NanoPhotometer^®^ spectrophotometer (IMPLEN, Westlake Village, CA, USA). Samples had an RNA Integrity Number (RIN) ≥ 7.0. The quality control was assessed with the Agilent 2100 bioanalyzer (Agilent Technologies, Santa Clara, CA, USA). The mRNA with a polyadenylic acid tail was enriched by connecting oligothymidine magnetic beads, and then the obtained mRNA was randomly interrupted with divalent cations in NEB fragmentation buffer. DNA library generation and RNA-seq high-throughput sequencing were performed by Shenzhen Microeco Biotech (Shenzhen, China). A Qubit2.0 Fluorometer was used for preliminary quantification; the library was diluted to 1.5 ng/uL, and then the Agilent 2100 bioanalyzer was used to detect the insert size of the library. Clustering of the index-coded samples was performed on a cBot Cluster Generation System using a TruSeq PE Cluster Kit v3-cBot-HS according to the manufacturer’s instructions. A total of 8 qualified libraries were sequenced on the Illumina Novaseq platform HiSeqTM 2500 (Illumina, San Diego, CA, USA), and 150 bp paired-end reads were generated.

### 4.3. Preprocessing of RNA-Seq Data

Raw sequence reads were processed using the NGSQC Toolkit (National Institute of Plant Genome Research, New Delhi, India, http://www.nipgr.res.in/ngsqctoolkit.html, accessed on 5 December 2020) to remove reads containing poly-N, low-quality or adaptor-polluted reads. The clean reads were mapped to a B73 reference maize genome [50] (version4, Cold Spring Harbor Laboratory, New York, NY, USA, ftp://ftp.gramene.org/pub/gramene/release-63/fasta/zea_mays/Zea_mays.B73_RefGen_v4.dna.toplevel.fa.gz, accessed on 5 December 2020) using hisat2 [51] (University of Texas Southwestern Medical Center, Dallas, TX, USA, https://ccb.jhu.edu/software/hisat2/index.shtml, on 5 December 2020) with default parameters. The number of fragments per kilobase of transcript per million mapped reads (FPKM) value for each gene was calculated using cufflinks [52] (version 2.2.1, Johns Hopkins University, Maryland, USA, accessed on 5 December 2020); only those genes with an FPKM value ≥1 in at least one stage were retained for further analysis. Gene expression profile clustering was performed using Mfuzz software [53] by http://www.bioinformatics.com.cn, an online platform for data analysis and visualization, accessed on 20 June 2021.

### 4.4. Identification of DEGs and Functional Classification

SizeFactors and nbinomTest functions of the DESeq2 R package (Bell Laboratories, Boston, MA, USA, http://bioconductor.org/biocLite.R, accessed on 5 December 2020) were performed to identify differentially expressed genes (DEGs) in the two samples. The threshold for defining significantly differentially expressed genes was as follows: adjusted *p*-value ≤ 0.05 and fold change >2 [54]. Clustering analysis of the DEGs between different growth stages is shown in a Venn diagram using Venn 2.1 (Juan Carlos Oliveros, Madrid, Spain, https://bioinfogp.cnb.csic.es/tools/venny/, accessed on 19 June 2021). The hierarchical cluster analysis output and heatmap were generated using TBtools [55] (South China Agricultural Uinversity, Guangzhou, China, https://github.com/CJ-Chen/TBtools/releases, accessed on 20 June 2021). Gene Ontology (GO) enrichment analysis was performed according to Ashburner et al. [56]. A hypergeometric test was used to identify significantly enriched GO terms with *p*-values ≤ 0.05. Pathway enrichment analysis of the Kyoto Encyclopedia of Genes and Genomes (KEGG) was performed as per Kanehisa and Goto [57]. Pathways for which the *p*-values were ≤0.05 were defined as significant enrichment. All these analyses were carried out using the online OmicShare tools (GENE DENOVO, Guangzhou, China, http://www.omicshare.com/tools, accessed on 20 June 2021).

### 4.5. Real-Time PCR Analysis

Total RNA was extracted from the shanks at four developing stages (L1–L4). The first strand of cDNA was synthesized according to the instructions for the FastQuant RT Kit (TIANGEN, Beijing, China). Subsequently, qRT-PCR was performed using the SYBR PrimeScript RT-PCR Kit (TakaRa, Dalian, China) with SYBR Green dye. The maize actin gene *ZmActin* was used as the internal control. The 2^–^^ΔΔCT^ [58] quantitative analysis method was used to calculate the relative expression level. The primers used in this study are listed in Appendix A.

## 5. Conclusions

In this study, the transcriptome sequencing of maize ear shanks at different developing stages was carried out. A total of 8145 DEGs were identified, presenting a dynamic view of transcriptome variation in maize shank development. Phytohormones (auxin, brassinosteroids, gibberellin, etc.), xyloglucan and xyloglucan xyloglucosyl transferase, and transcription factor families may play a crucial role in maize shank elongation processes. The results of this study may help to develop the understanding of the genetic architecture of ear shank elongation, which may be useful for breeding maize varieties with a high dehydration rate, and to reduce maize production costs through mechanized harvesting.

## Figures and Tables

**Figure 1 ijms-22-07029-f001:**
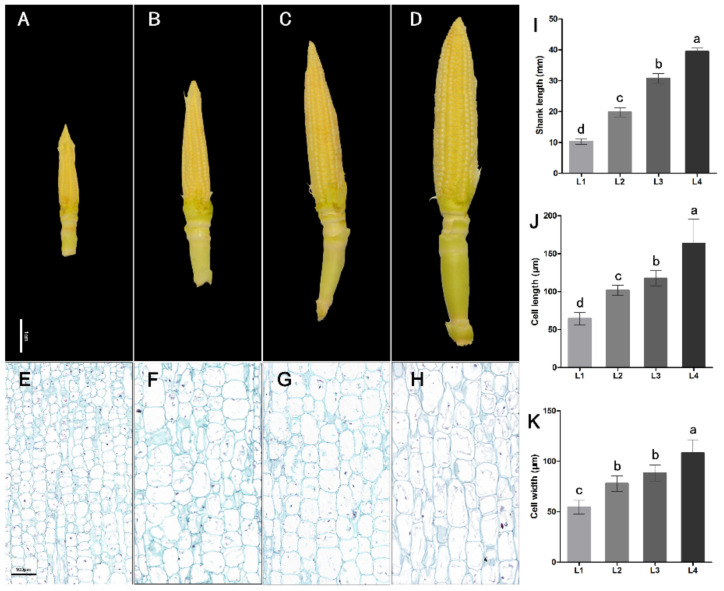
Change in the shank during the development process of maize ears. (**A**–**D**) Maize ear shanks with different lengths (L1~L4); (**E**–**H**) hand-cut longitudinal section between the second nodes of the four stages in the 0.5 × 1 mm area; (**I**) maize ear shank length from L1 to L4; (**J**) average cell length in different developing processes; (**K**) average cell width from L1 to L4. Lowercase letters represent level of significance.

**Figure 2 ijms-22-07029-f002:**
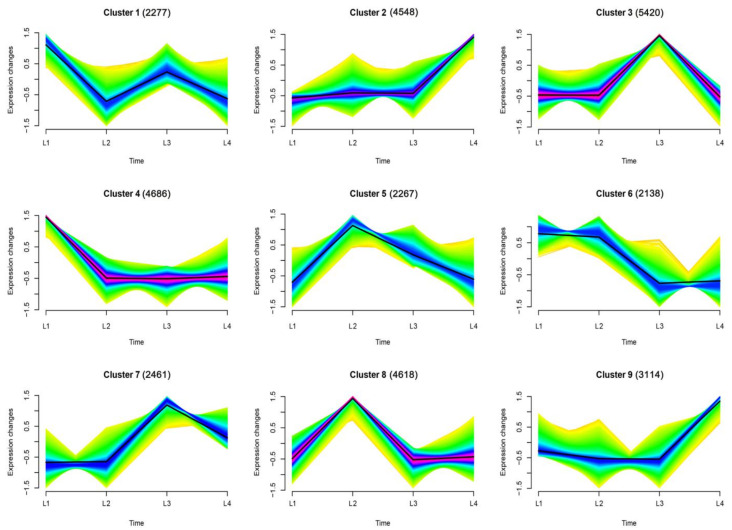
Gene expression dynamics of the maize ear shank elongation process. Nine clusters were obtained using Mfuzz clustering log_2_-fold change data of L1–L4 stages. Color changes (red–blue–green) represent the coincidence degree of a gene change with central variation in the cluster. Green indicates a low degree of coincidence, and red indicates a high degree of coincidence. The number of genes is indicated above each graph.

**Figure 3 ijms-22-07029-f003:**
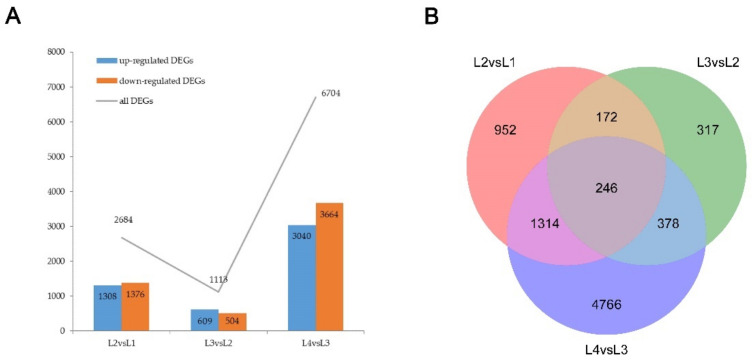
Numbers of differentially expressed genes (DEGs) in different comparison groups during the shank elongation process. (**A**) Numbers of up- and down-regulated DEGs in different comparison groups. (**B**) Venn diagram for unique DEGs in different comparison groups.

**Figure 4 ijms-22-07029-f004:**
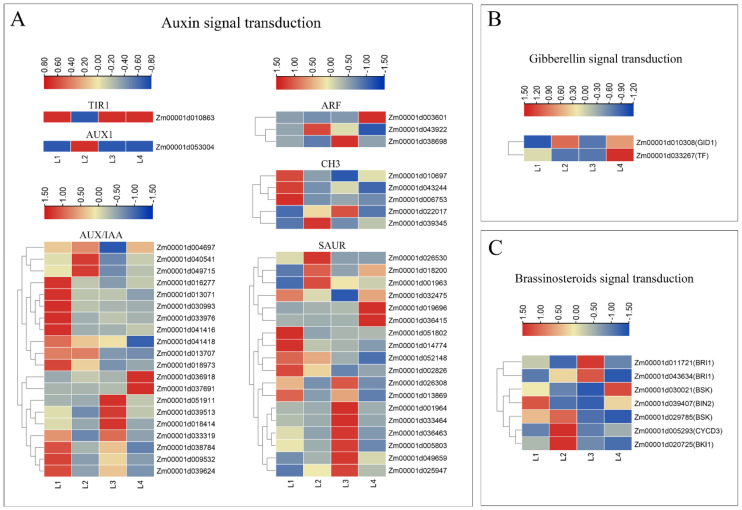
Expression of DEGs involved in “Plant hormone signal transduction”. (**A**) Heatmaps of DEGs which were enriched in auxin signal transduction. (**B**) Heatmaps of DEGs which were enriched in gibberellin signal transduction. (**C**) Heatmaps of DEGs which were enriched in brassinosteroid signal transduction.

**Figure 5 ijms-22-07029-f005:**
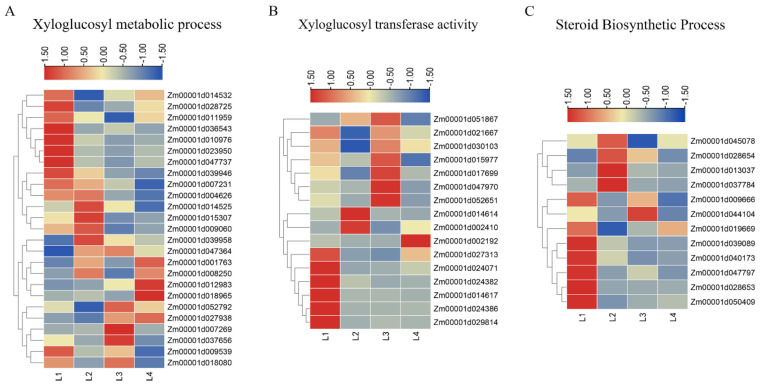
Expressions of DEGs were enriched in the xyloglucosyl metabolic process and xyloglucosyl transferase activity. (**A**) Heatmaps of DEGs were enriched in the xyloglucan metabolic process. (**B**) Heatmaps of DEGs were enriched in xyloglucan xyloglucosyl transferase activity. (**C**) Heatmaps of DEGs were enriched in the steroid biosynthetic process.

**Figure 6 ijms-22-07029-f006:**
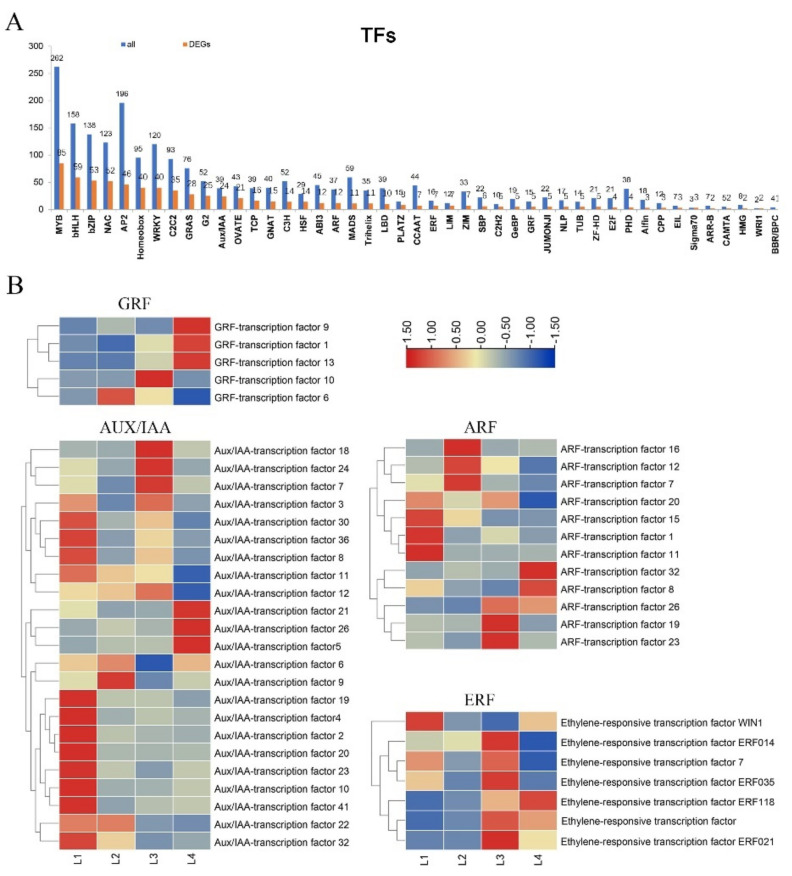
Expression of TFs involved in phytohormone response factor. (**A**) The distribution of transcription factor families with DEGs during maize ear shank elongation. (**B**) Heatmaps of the DEGs involved in hormone-related transcription factors during maize ear shank elongation.

**Figure 7 ijms-22-07029-f007:**
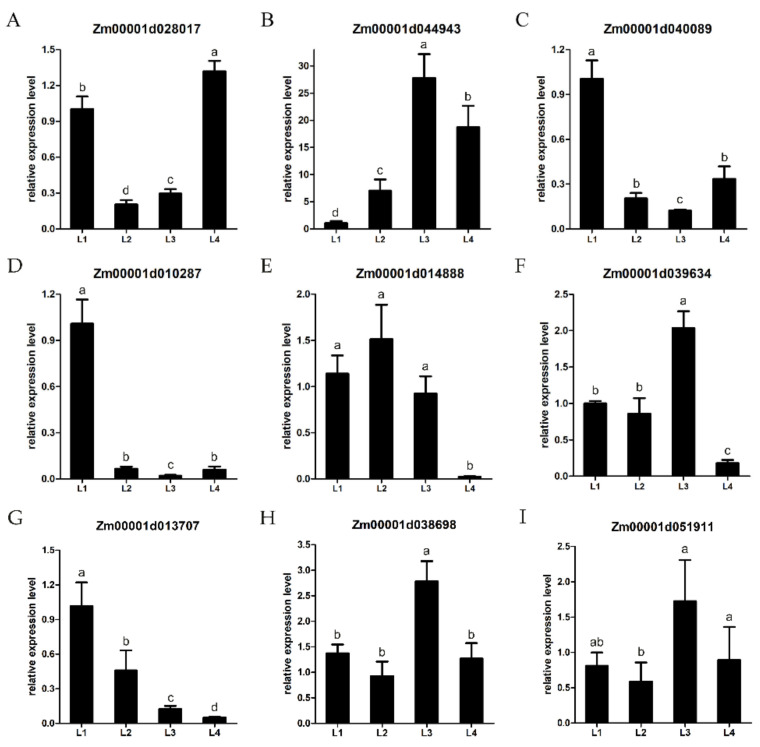
(**A**–**I**) Quantitative RT-PCR (qRT-PCR) validation of DEGs at different stages. Lowercase letters represent the level of significance.

**Figure 8 ijms-22-07029-f008:**
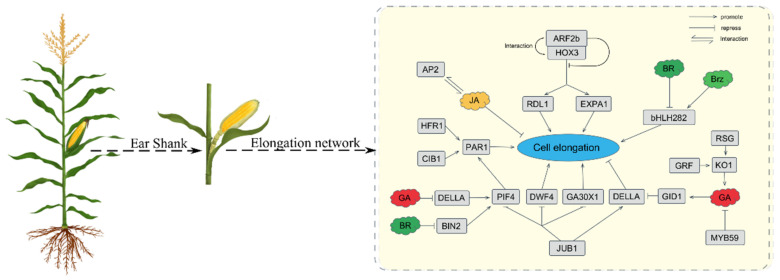
The effect of hormones and transcription factors on the elongation of maize shank cells.

**Table 1 ijms-22-07029-t001:** Top 20 GO terms.

GO ID	GO Term	Gene Number	*p*-Value	FDR
GO:0016798	hydrolase activity, acting on glycosyl bonds	157	6.00 × 10^−10^	3.95 × 10^−7^
GO:0004672	protein kinase activity	436	1.10 × 10^−9^	3.95 × 10^−7^
GO:0006468	protein phosphorylation	441	8.92 × 10^−9^	1.13 × 10^−5^
GO:0004553	hydrolase activity, hydrolyzing O-glycosyl compounds	143	9.34 × 10^−9^	2.23 × 10^−6^
GO:0015979	photosynthesis	57	1.99 × 10^−8^	1.26 × 10^−5^
GO:0008017	microtubule binding	50	4.64 × 10^−8^	8.32 × 10^−6^
GO:0015631	tubulin binding	50	6.95 × 10^−8^	9.64 × 10^−6^
GO:0016773	phosphotransferase activity, alcohol group as acceptor	463	8.06 × 10^−8^	9.64 × 10^−6^
GO:0007017	microtubule-based process	78	1.54 × 10^−7^	6.46 × 10^−5^
GO:0015630	microtubule cytoskeleton	65	2.36 × 10^−7^	2.05 × 10^−5^
GO:0009579	protein complex binding	50	3.19 × 10^−7^	3.27 × 10^−5^
GO:0044436	thylakoid part	35	3.20 × 10^−7^	2.05 × 10^−5^
GO:0032403	thylakoid	35	3.20 × 10^−7^	2.05 × 10^−5^
GO:0005871	kinesin complex	40	4.11 × 10^−7^	2.05 × 10^−5^
GO:0034357	photosynthetic membrane	34	4.41 × 10^−7^	2.05 × 10^−5^
GO:0009521	photosystem	33	6.06 × 10^−7^	2.35 × 10^−5^
GO:0005875	microtubule associated complex	44	1.25 × 10^−6^	4.15 × 10^−5^
GO:0016301	kinase activity	461	1.29 × 10^−6^	1.16 × 10^−4^
GO:0016310	phosphorylation	459	1.52 × 10^−6^	4.80 × 10^−4^
GO:0043228	intracellular non-membrane-bounded organelle	257	2.34 × 10^−6^	6.07 × 10^−5^

**Table 2 ijms-22-07029-t002:** Top 20 enriched KEGG pathways.

Pathway ID	Pathway	Out (1257)	All (4604)	*p*-Value
ko03010	Ribosome	151	375	9.38 × 10^−9^
ko01110	Biosynthesis of secondary metabolites	341	1023	7.80 × 10^−7^
ko00062	Fatty acid elongation	25	39	1.56 × 10^−6^
ko01100	Metabolic pathways	565	1819	2.38 × 10^−6^
ko00061	Fatty acid biosynthesis	23	39	3.01 × 10^−5^
ko00195	Photosynthesis	37	76	5.09 × 10^−5^
ko00196	Photosynthesis—antenna proteins	16	24	6.30 × 10^−5^
ko01212	Fatty acid metabolism	31	61	7.37 × 10^−5^
ko04626	Plant–pathogen interaction	126	348	9.81 × 10^−5^
ko00520	Amino sugar and nucleotide sugar metabolism	49	116	3.14 × 10^−4^
ko04075	Plant hormone signal transduction	96	263	4.93 × 10^−4^
ko00940	Phenylpropanoid biosynthesis	72	190	7.54 × 10^−4^
ko00360	Phenylalanine metabolism	20	39	1.21 × 10^−3^
ko00941	Flavonoid biosynthesis	16	29	1.36 × 10^−3^
ko00130	Ubiquinone and other terpenoid-quinone biosynthesis	19	37	1.56 × 10^−3^
ko00591	Linoleic acid metabolism	9	13	1.94 × 10^−3^
ko03410	Base excision repair	16	34	1.06 × 10^−2^
ko00250	Alanine, aspartate and glutamate metabolism	19	45	2.14 × 10^−2^
ko00780	Biotin metabolism	7	13	3.85 × 10^−2^
ko00760	Nicotinate and nicotinamide metabolism	8	16	4.44 × 10^−2^

## Data Availability

All the sequencing data of this study are available in the National Center for Biotechnology Information Sequence Read Archive database, with the accession number PRJNA738962 (https://www.ncbi.nlm.nih.gov/sra/PRJNA738962, accessed on 17 June 2021).

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
