# Peer review of "Comparative Transcriptome Analysis Reveals Regulatory Networks during the Maize Ear Shank Elongation Process"

_ijms, 2021, doi:10.3390/ijms22137029_

Round 1
Reviewer 1 Report
To investigate the genetic basis of maize ear shank elongation process, the authors performed RNA-seq analysis on maize shank elongation at four different developmental stages. Then conventional routine analyses were subsequently done on the RNA-seq data, such as clustering, GO and pathways analysis, transcription factor discovery, etc. The goal was to find some key genes involved in maize shank elongation.
While the research topic is interesting and the manuscript is well organized, there are some major issues in the analyses that need to be ruled out.
- The authors should try to make the article concise by presenting key results, instead of piling up all the analyses. For example, Table 1 is not necessary in the main text and can be moved into the supporting materials.
- One of the biggest issues in the analyses is that the authors did not do the necessary preprocessing before the clustering analysis. Many genes can be just low expressed in all the four stages. The authors should use some tools like ANOVA to identify those genes that are really differentially expressed in different stages and then perform clustering on such genes. Otherwise, the noisy genes can blur the clustering patterns a lot. Similarly, before doing differential gene analysis, the authors should remove those genes that have very low expression abundance.
- The authors didn't mention how GO classification analysis and KEGG pathway analysis were done and what tools they used. This doesn't make sense since GO and pathway analyses are one of the main content of the manuscript. The authors didn't mention which database was used to retrieve the information on maize transcription factors either.
- The data availability is not clear. The authors should deposit their RNA-seq data into a public database such as GEO for other researchers to reproduce the results.
- Some wet-bench experiments on validating the key genes will strengthen the work a lot.
- Language editing is needed to remove the English grammars, such as " significantly length"," corresponding for", " significantly difference".
Author Response
To investigate the genetic basis of maize ear shank elongation process, the authors performed RNA-seq analysis on maize shank elongation at four different developmental stages. Then conventional routine analyses were subsequently done on the RNA-seq data, such as clustering, GO and pathways analysis, transcription factor discovery, etc. The goal was to find some key genes involved in maize shank elongation.
While the research topic is interesting and the manuscript is well organized, there are some major issues in the analyses that need to be ruled out.
1. The authors should try to make the article concise by presenting key results, instead of piling up all the analyses. For example, Table 1 is not necessary in the main text and can be moved into the supporting materials.
>Thanks for your good evaluation and kind suggestion.
In this version, Table 1 has been moved into the supporting materials(Table S1).
2. One of the biggest issues in the analyses is that the authors did not do the necessary preprocessing before the clustering analysis. Many genes can be just low expressed in all the four stages. The authors should use some tools like ANOVA to identify those genes that are really differentially expressed in different stages and then perform clustering on such genes. Otherwise, the noisy genes can blur the clustering patterns a lot. Similarly, before doing differential gene analysis, the authors should remove those genes that have very low expression abundance.
> Thanks for your good evaluation and kind suggestion.
Considering some genes may only expressed (or highly expressed) at some specific stage and refered to other articles(like: Transcriptomic and physiological analyses of rice seedlings under different nitrogen supplies provide insight into the regulation involved in axillary bud outgrowth , Wang et al. BMC Plant Biology (2020) 20:197; Transcriptomic reprogramming in soybean seedlings under salt stress, Liu et al. Plant Cell & Environ. 2019;42:98–114). In this revised manuscript, we changed the analysis strategy. “only these genes with FPKM value ≥1 in at least one stage were retained for further analysis.” and removed these genes have lower expression levels (<1) in all four developing stages.
3.1 The authors didn't mention how GO classification analysis and KEGG pathway analysis were done and what tools they used. This doesn't make sense since GO and pathway analyses are one of the main content of the manuscript.
> Thanks for your suggestions.
In our study, we described the menthod to perform GO classification analysis and KEGG pathway analysis in “Materials and Method” section as follows:
Gene Ontology (GO) enrichment analysis was performed according to Ashburner et al. [57]. A hypergeometric test was used to identify significantly enriched GO terms with p-values ≤ 0.05. Pathway enrichment analysis of the Kyoto Encyclopedia of Genes and Genomes (KEGG) was performed as per Kanehisa and Goto [58]. Pathways for which the p-values were ≤0.05 were defined as significant enrichment. All these analyses were carried out using the online OmicShare tools (http://www.omicshare.com/tools).
3.2 The authors didn't mention which database was used to retrieve the information on maize transcription factors either.
>Thanks for your suggestions. In this study, transcription factors was obtained from Plant Transcription Factor Database (PlantTFDB, http://planttfdb.cbi.pku.edu.cn/) was added in 2.7 section as follows:
Here, maize TF gene family information was obtained from the Plant Transcription Factor Database (PlantTFDB 2.0, http://planttfdb.cbi.pku.edu.cn/).
4. The data availability is not clear. The authors should deposit their RNA-seq data into a public database such as GEO for other researchers to reproduce the results.
> Thanks a lot! We are very sorry for our negligence.
In this revised version, “Data Availability Statement” section was added at the back of the manuscript.
Data Availability Statement: All the squencing data of this study are available in ths National Center for Biotechnology Information Sequence Read Archive database with the accession number PRJNA738962 (https://www.ncbi.nlm.nih.gov/sra/PRJNA738962).
5. Some wet-bench experiments on validating the key genes will strengthen the work a lot.
>Thanks for your advice.
In this revised version, some key genes such as such as Aux/IAA-transcription factor 22(Zm00001d013707), ARF-transcription factor 23 (Zm00001d038698) and Aux/IAA-transcription factor 10(Zm00001d051911) were randomly selected to validate the RNA-seq results.
6. Language editing is needed to remove the English grammars, such as " significantly length"," corresponding for", " significantly difference".
> Thanks for your suggestion. We have used MDPI English editing service to check grammar, spelling, punctuation and some improvement of style in the revised version.
Those revised words and sections are marked in red.

Reviewer 2 Report
Authors identified the differentially expressed genes via transcriptome
sequencing analysis during ear shank elongation process, analyzing metabolic pathways. The study is interesting and important for the readers involved in the maize genetics and the results could be useful for geneticist and breeders of maize. I have only one concern, author used abbreviation in the result section, which were not explained before. Plesea use the full terms when used first time and later after abreviate them
Author Response
Authors identified the differentially expressed genes via transcriptome sequencing analysis during ear shank elongation process, analyzing metabolic pathways. The study is interesting and important for the readers involved in the maize genetics and the results could be useful for geneticist and breeders of maize. I have only one concern, author used abbreviation in the result section, which were not explained before. Plesea use the full terms when used first time and later after abreviate them.
>Thanks for your good evaluation and kind suggestion.
In this revised version, All the full terms of abbreviations were added when used first time in the manuscript, and we also added the “Abbreviations” sction at the end of our manuscript.
We also using MDPI English editing service to check grammar, spelling, punctuation and some improvement of style in the revised version.
Those revised words and sections are marked in red.

Round 2
Reviewer 1 Report
The authors have successfully addressed my concerns.